# MULTI-TASK LEARNING FOR DOCUMENT RANKING AND QUERY SUGGESTION

**Wasi Uddin Ahmad & Kai-Wei Chang**
Department of Computer Science
University of California, Los Angeles
{wasiahmad,kwchang}@cs.ucla.edu

**Hongning Wang**
Department of Computer Science
University of Virginia
hw5x@virginia.edu

## ABSTRACT

We propose a *multi-task* learning framework to jointly learn document ranking and query suggestion for web search. It consists of two major components, a document ranker and a query recommender. Document ranker combines current query and session information and compares the combined representation with document representation to rank the documents. Query recommender tracks users' query reformulation sequence considering all previous in-session queries using a sequence to sequence approach. As both tasks are driven by the users' underlying search intent, we perform joint learning of these two components through session recurrence, which encodes search context and intent. Extensive comparisons against state-of-the-art document ranking and query suggestion algorithms are performed on the public AOL search log, and the promising results endorse the effectiveness of the joint learning framework.

## 1 INTRODUCTION

Understanding users' information need is the key to optimize a search engine for providing relevant search results. Search engine logs have been extensively used to mine users' search intent, reflected in their search result click preferences and query reformulations (Baeza-Yates et al., 2004; Croft et al., 2010). Typically, user query logs are partitioned into search sessions, i.e., sequences of queries and clicks issued by the same user and within a short time interval. Search sessions provide useful contextual information about user intent and help to narrow down ambiguity while ranking documents for the current query and predicting next query that users will submit, a.k.a. *context-awareness* (Jiang et al., 2014). Since both a user's click behavior and query reformulation are driven by the underlying search intent, we argue that jointly modeling both tasks can benefit each other.

In this work, we propose a joint learning framework, called *multi-task neural session relevance framework* (M-NSRF), to predict users' result clicks and future queries in search sessions. We model search context within a session via a recurrent latent state in a deep neural network (Collobert & Weston, 2008; Liu et al., 2015), which governs the generation of result clicks in the current query and formation of next query. By sharing the latent states across the tasks of document ranking and query suggestion, we learn the representations of queries, documents and user intent carrying over the whole session jointly, i.e., multi-task learning. This multi-task learning framework is flexible and can be incorporated with existing approaches for representing query and documents.

The general workflow of M-NSRF is illustrated in Figure 1. Given a sequence of queries from the same search session, e.g., *"cheap furniture"* and *"craig list virginia"*, M-NSRF is trained to predict both the result clicks under the current query and the next query *"cheap furniture for sale."* It is evident that in this search session the user kept reformulating the queries because his/her information need has not been met by the result clicks in the previously submitted queries, reflected in the added and removed query terms. And such revisions suggest what he/she might want to click next. As a result, the modeling of result clicks and query reformulations mutually reinforce each other to reveal users' underlying search intent. In M-NSRF, we model user search intent as a session-level recurrent state, the learning of which is aided by both document ranking and query prediction tasks.

We evaluate the effectiveness of the proposed framework using the publicly available AOL search log and compare with several well-known classical retrieval models, as well as various neural retrieval models specifically designed for ad-hoc retrieval. We also compare M-NSRF with several

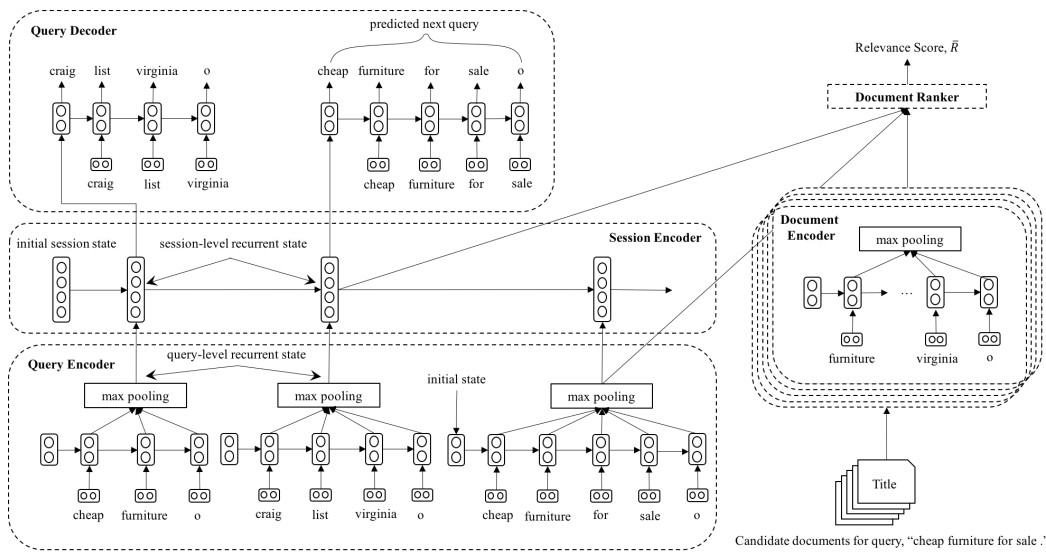

Figure 1: General workflow of the proposed multi-task neural session relevance framework. The framework is trained on search sessions to jointly predict next query and rank corresponding documents. The model encodes the current query in the session, *"craig list virginia"*, updates corresponding session-level recurrent state, maximizes the probability of the next query, *"cheap furniture for sale"* (i.e., the task of query suggestion), encodes the candidate documents for the following query and minimizes loss for clicked documents (i.e., the task of document ranking).

baseline models for the query prediction task. The empirical results show that by leveraging information in both tasks, the proposed approach outperforms existing models significantly on ad-hoc document retrieval task and exhibits competitive performance in query suggestion.

To summarize, the key contributions of this work include:

1. We propose a novel multi-task neural session relevance model that is jointly trained on document ranking and query suggestion tasks by utilizing in-session queries and clicks in a holistic way.

2. We provide detailed experiment analysis, and release the implementation[1] and the data processing tool to facilitate future research.

## 2    RELATED WORK AND BACKGROUND

**Ad-hoc Retrieval.** Traditional retrieval models such as query likelihood (Ponte & Croft, 1998) and BM25 (Robertson et al., 2009) are based on exact matching of query and document words with a variety of smoothing, weighting and normalization techniques. Recently, deep neural network based approaches demonstrate strong advance in ad-hoc retrieval. Existing neural ranking models fall into two categories: representation focused (Huang et al., 2013; Gao et al., 2014) and interaction focused (Guo et al., 2016b). The earlier focus of neural ranking models was mainly on representation based models (Hu et al., 2014; Shen et al., 2014), in which the query and documents are first embedded into continuous vectors, and the ranking is calculated based on their embeddings' similarity. The interaction focused neural models (Hu et al., 2014; Pang et al., 2016; Guo et al., 2016a), on the other hand, learn query-document matching patterns from word-level interactions. Both the interaction and representation focused models can be combined for further improvements (Mitra et al., 2017). Similarly, Jaech et al. (2017) captures both local relevance matching and global topicality signals when computing relevance of a document to a query. Our work falls into the representation focused approach to form query and document representations and jointly models the two tasks through session representation learning.

**Query Suggestion.** In general, query suggestion algorithms define various distance metrics between queries to find the most similar ones as suggestions for one another (Wen et al., 2001; Baeza-Yates

---

[1]https://github.com/wasiahmad/mnsrf_ranking_suggestion

et al., 2004). The closest work to ours is the end-to-end hierarchical recurrent encoder-decoder architecture (HRED-qs) (Sordoni et al., 2015), which ranks candidates for context-aware query suggestion. Our proposed framework differs from HRED-qs as it integrates another important type of user search behavior, i.e., result clicks, into the joint modeling, which provides additional contextual information for query suggestion. Mitra & Craswell (2015) proposed a candidate generation approach for rare prefixes using frequently observed query suffixes and suggested a neural model to generate ranking features along with n-gram features. Similarly, a large pool of previous works investigated task or context-aware approaches (Cao et al., 2008; He et al., 2009; Feild & Allan, 2013; Chen et al., 2017; Garigliotti & Balog, 2017) for query suggestion.

**Multi-task Learning.** The goal of multi-task learning is to improve generalization on the target task by leveraging the domain-specific information contained in the training signals of related tasks (Caruana, 1998). Multi-task learning in combination with deep neural networks has been successfully used in many application scenarios, including natural language processing (Collobert & Weston, 2008; Liu et al., 2016a;b; Peng & Dredze, 2017; Peng et al., 2017), speech recognition (Deng et al., 2013; Thanda & Venkatesan, 2017) and computer vision (Girshick, 2015). However, it has been less explored in the information retrieval domain. Liu et al. (2015) proposed a multi-task deep neural approach to combine query classification and document ranking, and reported improvement on both the tasks. Bai et al. (2009) used multi-task learning in learning to rank for web search. We propose to jointly learn document ranking and query suggestion via multi-task learning to better capture latent intent embedded in users' search behaviors.

## 3 MULTI-TASK NEURAL SESSION RELEVANCE FRAMEWORK

We define a session to be a sequence of queries, $Q = \{Q_1, \ldots, Q_n\}$, submitted by a user in a short time interval in a chronological order to satisfy a specific search intent. Every query $Q_i$ in a session is associated with a set of related documents $D = \{D_1, \ldots, D_m\}$ which need to be ranked by its relevance to the query, and $o = \{o_1, \ldots, o_m\}$ is the set of relevance labels for each document in $D$. In typical search engine logs, $o_j$ is usually approximated via user clicks, e.g., $o_j = 1$ if and only if the document $D_j$ is clicked. A query $Q_i$ and a document $D_j$ consist of a sequence of words, i.e., $Q_i = \{w_i^1, \ldots, w_i^q\}$ and $D_j = \{w_j^1, \ldots, w_j^d\}$, where $q = |Q_i|$ and $d = |D_j|$ are the query and document lengths respectively. $V$ is the size of vocabulary constructed over queries and relevant documents.

The document ranking refers to the task of ordering the retrieved results with respect to their relevance to the given query under the users' search intent. Accurate modeling of relevance between a document and a query is the key in this task. In this work, we model ranking of related documents as a pointwise classification problem where the goal is to predict the relevance label between a query and a document. But our developed solution can be extended to pairwise or list-wise ranking models (Liu et al., 2009). On the other hand, query suggestion refers to the task of predicting next query $Q_i$ based on previous queries $Q_1 \ldots Q_{i-1}$ by users in the same session, so as to help them explore the search space. The key challenge is to maintain the semantic consistency between the suggested queries and users' original queries, with respect to their search intent.

Traditional IR approaches consider document ranking and query suggestion tasks separately (Huang et al., 2013; Shen et al., 2014; Sordoni et al., 2015), although both the tasks are driven by users' underlying search intent. In contrast, M-NSRF is designed to jointly learn a document ranker and a query recommender by modeling shared search context embedding in a session. Based on the latent session state inferred from all the previous queries and clicks in the same session, the *document ranker* is trained to predict user clicks from the candidate documents for the current query and the *query recommender* is trained in a sequence to sequence fashion (Sutskever et al., 2014) to predict the user's next query. The process is repeated sequentially for all the queries in the same session. The detailed architecture of the proposed multi-task neural session relevance framework (M-NSRF) is provided in Appendix A. In the following, we discuss each component of M-NSRF.

### 3.1 DOCUMENT RANKER

Ranking the retrieved documents for an input query requires encoding the query and documents into a shared representation space. In addition to the search intent carried by the current query, search context, which is reflected in the previously submitted queries in the same session, should also be accounted in ranking the documents. In M-NSRF, we model the latent user search intent

in a sequence of queries via a series of session recurrent states. As a result, the document ranker component in M-NSRF consists of a query encoder, a document encoder, a session encoder and a ranker. The ranker sub-component combines the latent representations of query and session and match them with document representations to generate the ranking score of documents, based on which the documents are ordered. The technical details of each constituent element are given as follows.

**Query Encoder**. The query encoder encodes a query into a vector. To encode a sequence of words, various approaches have been studied. We follow (Conneau et al., 2017) to adopt a bidirectional LSTM with max pooling (BiLSTM-max), due to its superior practical performance. Considering query as a sequence of words $Q_i = \{w_i^1, \ldots, w_i^q\}$, the encoder composed of forward and backward LSTM reads the sequence in two opposite directions,

$$\overrightarrow{h}_t = LSTM_t(\overrightarrow{h}_{t-1}, w_i^t), \quad \overleftarrow{h}_t = LSTM_t(\overleftarrow{h}_{t+1}, w_i^t), \quad h_t = [\overrightarrow{h}_t, \overleftarrow{h}_t]$$

where $h_t \in R^{2d}$ is the query-level recurrent state, $d$ is the dimensionality of the LSTM hidden unit initialized to a zero vector. To form a fixed-size vector representation of variable length queries, maximum value is selected over each dimension of the hidden units,

$$Q_{i,k} = \max_q h_{k,q}, \; k = 1, \ldots, d$$

where $Q_{i,k}$ is the $k$-th element of the latent vector $Q_i$.

**Document Encoder**. The goal of the document encoder is to encode documents into continuous vectors. We use the same BiLSTM-max technique that is utilized in encoding queries as the document encoder. The only difference is in the dimensionality of the LSTM hidden units. In general, because a document (body or title) is longer than a query, we consider dense vector of a larger size as the continuous representations of the documents.

**Session Encoder**. The session encoder generates a representation to encode the queries that the system has received so far. Unlike query and document encoders that can read the complete encoded queries and documents, the session encoder does not have information from the future queries. Therefore, we use a unidirectional LSTM (Hochreiter & Schmidhuber, 1997) for session encoding. The session encoder takes the sequence of query representations $Q_1, ..., Q_n$ as input and computes the sequence of session-level recurrent states.

$$S_i = LSTM_i(S_{i-1}, Q_i),$$

where $S_i \in R^d$ is the session-level recurrent state initialized to a zero vector. As a result, each query in the session has its session-level recurrent state, summarizing the user's information need that has been processed up to query $Q_i$.

**Ranker**. We first concatenate the current query representation $Q_i$ with previous session-level recurrent state $S_{i-1}$ via a non-linear transformation. This combined representation reflects the search intent reflected in the current query and the past ones in the same session. Then we compute the ranking score of document $D_j$ under the query as its probability of being relevant via a sigmoid function (with binary relevance labels),

$$P(D_j|Q_i, S_{i-1}) = \sigma\big(D_j^T \tanh(W_r[Q_i, S_{i-1}] + b_r)\big), \; j = 1, \ldots, m \tag{1}$$

where $W_r \in R^{(d_q+d_s) \times d_d}$, $b_r \in R^{d_d}$, and $d_q$, $d_s$ and $d_d$ are the dimensionality of the query encoder, session encoder and document encoder hidden units and $\sigma$ is the sigmoid function. A list of retrieved documents can therefore be ordered by this ranking score.

## 3.2 QUERY RECOMMENDER

Query recommender suggests related queries to users by inferring the underlying search intent through utilizing in-session previous queries embedded in latent vectors. Following Sutskever et al. (2014) and Bahdanau et al. (2015), the query recommender in M-NSRF predicts users' next query in a sequence to sequence manner. Basically the query recommender module estimates the probability of the next query $Q_i = \{w_i^1, \ldots, w_i^q\}$, given all the previous queries up to position $i-1$ in a session as follows,

$$P(Q_i|Q_{1:i-1}) = \prod_{t=1}^q P(w_i^t|w_i^{1:t-1}, Q_{1:i-1})$$

We use LSTM as a basic building block for the query recommender. Information about all the previous queries represented through a session vector $S_i$ is passed to the query recommender. To

this end, the recurrent state of the query recommender is initialized with a non-linear transformation of $S_i$, $h_0 = \tanh(W_q S_i + b_q)$, where $h_0 \in R^d$ is the initial recurrent state. Then the query recommender's recurrence is computed by $h_t = LSTM_t(h_{t-1}, w_i^t)$, where $h_{t-1}$ is the previous hidden state, $w_i^{t-1}$ is the previous query term. Finally, each recurrent state is mapped to a probability distribution over the vocabulary of size $V$ using a combination of linear transformation and the softmax function. Word with the highest probability is chosen as the next word in sequence.

$$P(w|w_i^{1:t-1}, Q_{1:i-1}) = g(W_p h_t + b), \tag{2}$$

where $g$ is the softmax function that outputs a vector to represent the distribution of next words, where the probability assigned to the $j$-th element is defined as $g(z)_j = \frac{e^{z_j}}{\sum_{k=1}^{K} e^{z_k}}$, $j = 1, ..., K$.

**Query Suggestion**. In the decoding phase, similar to (Sordoni et al., 2015), we use greedy decoding algorithm for suggesting next query. Given a sequence of queries up to position $i - 1$, a suggested query $Q_i$ is:

$$Q^* = \arg\max_{Q' \in \mathcal{Q}} P(Q'|Q_{1:i-1})$$

where $\mathcal{Q}$ is the space of all possible queries. To generate $Q^* = \{w^1, \ldots, w^q\}$, we use a greedy approach like in (Sordoni et al., 2015) where $w^t = \arg\max_w P(w|w^{1:t-1}, Q_{1:i-1})$ To provide query suggestions of variable lengths, we use standard word-level decoding techniques. We iteratively consider the best prefix $w_{1:t}$ up to length $t$ and extend it by sampling the most probable word given the distribution in Eq. (2). The process ends when we obtain a well-formed query containing the special end-of-query token.

### 3.3 LEARNING END-TO-END

Within a session, M-NSRF ranks documents in a set of candidates and predicts next query given the current query sequence. Therefore, the training objective of M-NSRF consists of two terms. The first term is the binary cross entropy loss from the document ranker,

$$\mathcal{L}_1 \equiv -\frac{1}{m} \sum_j^m o_j \times \log P(D_j|Q_i) + (1 - o_j) \times \log(1 - P(D_j|Q_i))$$

where $o_j$ represents binary click label for $D_j$. The second term is the regularized negative log-likelihood loss from the query suggestion model,

$$\mathcal{L}_2 \equiv -\sum_t^q \log P(w_i^t|w_i^{1:i-1}, Q_{1:i-1}) + L_R, \tag{3}$$

where $\mathcal{L}_R \equiv -\lambda \sum_{w \in V} P(w|w_i^{1:t-1}, Q_{1:i-1}) \log P(w|w_i^{1:t-1}, Q_{1:i-1})$ is the regularization term to avoid the distribution of words in Eq. (2) from being highly skewed, and $\lambda$ is a hyper-parameter to control the regularization term. The final objective is the summation of $L_1$ and $L_2$ over all the queries.[2]

Note that the document ranker and query recommender share the same document, query, and session encoders, and the training of M-NSRF can be done in an online manner using the following procedure. In the forward pass, M-NSRF computes the query and corresponding document encodings, updates session-level recurrent states, click probability for each candidate document and the log-likelihood of each query in the session given the previous ones. In the backward pass, the gradients are computed and the parameters are updated based on the ADAM update rule (Kingma & Ba, 2014). Details of implementation can be found in Section 4.2

### 3.4 GENERALIZING M-NSRF FOR NEURAL IR MODELS

The proposed multi-task learning framework is general and the query encoder, document encoder, document ranker, and query recommender can be replaced by other designed architecture. In general, any neural query suggestion model working in the sequence to sequence fashion can be readily incorporated with the proposed multi-task learning framework. Similarly, most neural IR models that built on the notion of learning query and document representations in a latent space for relevance modeling can be Incorporated in our framework as well. However, due to distinctive nature of different neural IR models, careful study is required while adding context-awareness into the final architecture.

---

[2]We can also consider optimizing the weighted sum of $L_1$ and $L_2$. However, our preliminary experiments showed that giving them equal weights already performs well.

In this paper, we take the Match-Tensor model (Jaech et al., 2017), a recently proposed neural relevance model for document ranking as an example and extend it to a multi-task Match-Tensor (M-Match-Tensor) model by incorporating the query recommender component within our proposed multi-task learning framework. Different from NSRF, which embeds queries and documents into vectors, Match-Tensor learns a contextual representation for each word in the queries and documents. Therefore, documents and queries are represented as matrices. The document ranker in Match-Tensor then computes the relevant score based on the following formulation:

$$P(D_j|Q_i) = \sigma\big(W_r\, C(Q_i, D_j) + b_r\big),\ i = 1, \dots, m$$

where $C$ represents a sequence of convolutional operation (Lawrence et al., 1997) and max-pooling on the $2d$ product of the query and document vectors. The detailed architecture of the M-Match-Tensor model (M-Match-Tensor) is provided in Appendix B. To incorporate the match-tensor in our mulit-task learning framework, we can replace the document encoder, query encoder, and the document ranker in M-NSRF with the ones specified in the Match-Tensor model. However, as the computations involve in the document ranker is substantial, we do not increase the computational complexity further by adding session recurrence in the document ranking. The query decoding component in M-Match-Tensor is identical to M-NSRF. In the experiment, we will show that multi-task Match-Tensor achieves better performance than Match-Tensor in document ranking task.

## 4 EXPERIMENTS

### 4.1 DATA SETS AND EVALUATION METHODOLOGY

We conduct our experiments on the publicly available AOL search log (Pass et al., 2006). The queries in this dataset were sampled between 1 March, 2006 and 31 May, 2006. In total there are 16,946,938 queries submitted by 657,426 unique users. We removed all non-alphanumeric characters from the queries, applied word segmentation and lowercasing. We followed (Jansen Bernard et al., 2007) to define a session by a 30-minute window of inactive time, and filtered sessions by their lengths (minimum 2, maximum 10). We only kept the most frequent $|V| = 100k$ words and mapped all other words to an $<unk>$ token when constructing the vocabulary. We randomly selected 1,032,459 sessions for training, 129,053 sessions for development and 91,108 sessions for testing, with no overlapping. In total, there are 2,987,486 queries for training, 287,138 for development, and 259,117 for testing. The average length of the queries and documents (only the title field) are 3.15 and 6.77 respectively. In our experiments, we set maximum allowable length of query and document to 10 and 20 respectively.

In the document ranking task, we need to rank the most relevant (e.g., most clickable) document on top. We used three standard ranking metrics, mean average precision (MAP), mean reciprocal rank (MRR) and normalized discounted cumulative gain (NDCG) metric computed at positions one, three, five and ten, to measure the performance. Since AOL search log only contains clicked documents, we constructed the ranking candidates by the top ranked documents by BM25 (Robertson et al., 2009). Each query in the test set consists of 50 candidate documents including the clicked ones. However, to reduce the training time and memory use, each query in training and development set only contains 5 candidates.

In the query suggestion task, we need to suggest the most semantically related query to users. As we do not have user feedback on the suggested queries, we treated the users' next submitted query as ground-truth (Sordoni et al., 2015), and used the BLEU scores (Papineni et al., 2002) as the evaluation metric, which is a popularly used metric in machine translation and text generation tasks. In addition, following (Santos et al., 2013; Sordoni et al., 2015), we evaluated the query suggestion quality by mean reciprocal rank (MRR), i.e., to test if the algorithm can rank the users' next query on top of its recommendation list. In this set of experiments, given a set of candidate queries, query suggestion models give a likelihood score of generating the candidates. We follow Sordoni et al. (2015) for dataset split and generate candidates using a co-occurrence based suggestion model. Like Sordoni et al. (2015), we give the anchor queries (second last query of a session) as an input to the query suggestion model and evaluates the rank of the next query among the candidates.

### 4.2 BASELINES AND IMPLEMENTATION DETAILS

**Document Ranking Baselines**. We compared M-NSRF and M-Match Tensor with word-based baselines and neural network-based baselines. Word-based baselines include query likelihood model

Table 1: Comparison of document ranking models over the AOL search log.

| Model Type | Model Name | MAP | MRR | NDCG | | | |
| --- | --- | --- | --- | --- | --- | --- | --- |
| | | | | @1 | @3 | @5 | @10 |
| Traditional | BM25 | 0.164 | 0.172 | 0.121 | 0.136 | 0.141 | 0.156 |
| IR-models | QL | 0.139 | 0.146 | 0.088 | 0.108 | 0.122 | 0.133 |
| Embedding-based | ESM | 0.214 | 0.179 | 0.118 | 0.127 | 0.139 | 0.158 |
| Representation | DSSM | 0.263 | 0.287 | 0.152 | 0.206 | 0.248 | 0.315 |
| Focused | CLSM | 0.465 | 0.505 | 0.369 | 0.441 | 0.482 | 0.523 |
| | ARC-I | 0.383 | 0.413 | 0.238 | 0.343 | 0.404 | 0.467 |
| Interaction | DRMM | 0.277 | 0.316 | 0.221 | 0.242 | 0.267 | 0.304 |
| Focused | ARC-II | 0.423 | 0.455 | 0.294 | 0.386 | 0.442 | 0.501 |
| Representation and | DUET | 0.272 | 0.301 | 0.152 | 0.212 | 0.263 | 0.341 |
| Interaction Focused | Match-Tensor | 0.613 | 0.621 | 0.568 | 0.572 | 0.596 | 0.618 |
| Neural Session Model (this paper) | NSRF | 0.553 | 0.568 | 0.481 | 0.526 | 0.555 | 0.574 |
| Multi-task Model | M-NSRF | 0.581 | 0.603 | 0.523 | 0.568 | 0.583 | 0.614 |
| (this paper) | M-Match-Tensor | 0.621 | 0.634 | 0.572 | 0.578 | 0.602 | 0.632 |

based on Dirichlet smoothing (QL) (Ponte & Croft, 1998) and BM25 (Robertson et al., 2009). In addition, following (Mitra et al., 2016), we investigated the ranking performance of a simple word embedding-based model using GloVe word embeddings (Pennington et al., 2014). To compare with neural models, we consider baselines broadly categorized in representation-focused, interaction-focused and a combination of both. Representation-focused neural baselines include: DSSM (Huang et al., 2013), CLSM (Shen et al., 2014), ARC-I (Hu et al., 2014) and interaction-focused baselines include: ARC-II (Hu et al., 2014) and DRMM (Guo et al., 2016a). A combination of representation and interaction focused models include: DUET (Mitra et al., 2017) and Match Tensor (Jaech et al., 2017). Details of these models are provided in Appendix C. We implemented all the baseline models in PyTorch.

**Query Suggestion Baselines**. To evaluate the performance on query suggestion task, we consider three baseline methods including Seq2seq model proposed by Bahdanau et al. (2015), Seq2seq with global attention mechanism (Luong et al., 2015) and HRED-qs (Sordoni et al., 2015). Details of these models are provided in appendix D. We implemented all three baselines in PyTorch and optimized using negative log-likelihood loss as in Eq. (3).

**Implementation Details of M-NSRF**. The model was trained end-to-end and we used mini-batch SGD with Adam (Kingma & Ba, 2014) for optimization. with the two momentum parameters set to 0.9 and 0.999 respectively. We use 300-dimensional word vectors trained with GloVe (Pennington et al., 2014) on 840 billion of tokens to initialize the word embeddings. Out-of-vocabulary words were randomly initialized by sampling values from a zero-mean unit-variance normal distribution. All training used a mini-batch size of 32 to fit in single GPU memory. Learning rate was fixed to 0.001. We used dropout (0.20) (Srivastava et al., 2014) and early stopping with a patience of 5 epochs for regularization. We set $\lambda = 0.1$ for entropy regularization in Eq. (3). M-NSRF is implemented in PyTorch and it runs on a single GPU (TITAN X) with roughly a runtime of 90 minutes per epoch. In general, M-NSRF runs up to 20 epochs and we select the model that achieves the minimum loss on the development set.

### 4.3 EVALUATION RESULTS

**Document Ranking Quality**. Table 1 shows the performance of NSRF, M-NSRF, M-Match-Tensor and other baseline models. NSRF significantly outperforms all the baselines except the Match-Tensor model. However, the model size of NSRF is much smaller than Match-Tensor. With multi-task learning, both M-NSRF and M-Match-Tensor outperform NSRF and Match-Tensor, respectively. To study the advantage of multi-task learning on our proposed approach, we trained M-NSRF only on document ranking task (noted as NSRF in Table 1) and observed significant performance drop which endorses the mutual benefit of joint learning of these two tasks via multi-task learning.

Existing neural ranking models, like DRMM and DUET architecture, achieved sub-optimal performance in our experiments. We believe because of the simple architecture of DRMM with few hundreds of parameters, the model fell behind to show competitive performance on the evaluation

Table 2: Examples of next query suggested by M-NSRF given all previous queries in a session.

| | |
|---|---|
| Previous session queries | types of weapons of mass destruction, weapons of mass destruction, nuclear weapons |
| Next user query | biological weapons |
| Suggested next query | destructive nuclear weapons |
| Previous session queries | resume template, resume template free, resume word perfect template free, wordperfect com |
| Next user query | wordperfect resume templates free |
| Suggested next query | free microsoft word templates |

Table 3: Comparison of different query suggestion models.

| Model Name | BLEU | | | | MRR[2] |
|---|---|---|---|---|---|
| | 1 | 2 | 3 | 4 | |
| Seq2seq | 24.5 | 9.7 | 4.5 | 1.9 | 0.229 |
| Seq2seq with attention | 28.1 | 15.7 | 10.4 | 8.5 | 0.252 |
| HRED-qs | 26.4 | 13.6 | 7.9 | 5.8 | 0.231 |
| HRED-qs w/ entropy regularizer | 27.6 | 15.1 | 9.2 | 6.7 | 0.233 |
| M-NSRF | 26.8 | 14.1 | 8.4 | 6.1 | 0.235 |
| M-NSRF w/ entropy regularizer | 28.6 | 16.7 | 10.2 | 8.3 | 0.238 |

dataset. On the other hand, we believe that the use of smaller number of top character $n$-graphs (in our case, 5000) by the DUET architecture limits its effectiveness in modeling representation and interaction focused features to compute matching quality between query and document. We have to note that, some negative results have been reported for document title-based ad-hoc retrieval tasks (Guo et al., 2016a); and thus in our future work, we plan to investigate M-NSRF and all baseline models' performance with document body content considered.

**Query Suggestion Accuracy**. Examples of the predicted queries by M-NSRF given preceding queries from the same session is presented in Table 2 (more examples are provided in Appendix E). The quantitative comparison results in this task between our proposed framework and baseline models are presented in Table 3. While M-NSRF and HRED-qs consider information from all preceding queries in the same session, the other two baselines only consider the current query to predict the next one. Table 3 shows that M-NSRF outperformed seq2seq and HRED-qs baselines in all measured BLUE scores, and MRR; but the Seq2seq with attention baseline performed better than M-NSRF in terms of BLEU-3, BLEU-4 and MRR[3].

We investigated the advantage of attention mechanism in this particular task and found it performs well when there is a considerable overlap between the input and output queries. Specifically, in more than 25% of the test sessions, the input queries and their next queries are exactly the same; and the attention mechanism encourages the model to repeat words from the input query. When we restrict the experiment on test sessions that have no overlap between the input and its next query, the Seq2seq with attention baseline encountered significant performance drop, while M-NSRF provided improved performance compared to all baselines (approx. 5%, 10% and 1% improvement in terms of MRR over Seq2seq, Seq2seq with attention and HRED-qs model). The major reason for this improvement is that our model leverages the global session information and is therefore not restricted by the input query and is able to generate related but totally new queries (as shown in Appendix E).

We also observed that entropy-based regularization helped M-NSRF in predicting the next query, as shown in Table 3 that M-NSRF without regularization achieved worse performance in all measured BLUE scores. We further investigated the utility of regularization for the baselines. We found significant (~1.2%) improvement for the HRED-qs, but the performance improvement for the seq2seq and seq2seq with attention mechanism is rather marginal (~0.2%). However, the regularization technique does not introduce any significant difference to the ranking based evaluation.

---

[3] Note that Sordoni et al. (2015) used their neural models output as a feature for a learning-to-rank model. To understand the effectiveness of these neural models in query suggestion task, we directly compared their ranking quality. Therefore, the numbers reported in Table 3 are not comparable to that in Sordoni et al. (2015).

Table 4: Ablation study for performance analysis of M-NSRF. Statistical significances are compared with NSRF's full model and presented in bold-faced. † indicates M-NSRF is trained to learn word embeddings, no pre-trained embeddings were used.

| NSRF Variant | MAP | NDCG@1 | NDCG@3 | NDCG@10 |
|---|---|---|---|---|
| Full model | **0.581** | **0.523** | **0.568** | **0.614** |
| Fixed embeddings | 0.252 (−0.329) | 0.182 (−0.341) | 0.216 (−0.352) | 0.289 (−0.325) |
| Learned embeddings† | 0.302 (−0.279) | 0.222 (−0.301) | 0.261 (−0.307) | 0.297 (−0.317) |
| Mean-pool | 0.576 (−0.005) | 0.515 (−0.008) | 0.561 (−0.007) | 0.608 (−0.006) |
| BiLSTM-last | 0.563 (−0.018) | 0.505 (−0.018) | 0.541 (−0.027) | 0.594 (−0.020) |
| M-NRF | 0.553 (−0.028) | 0.494 (−0.029) | 0.544 (−0.024) | 0.582 (−0.032) |
| GloVe 6B 50d | 0.247 (−0.324) | 0.196 (−0.329) | 0.232 (−0.336) | 0.296 (−0.348) |
| GloVe 6B 100d | 0.312 (−0.269) | 0.241 (−0.282) | 0.273 (−0.295) | 0.306 (−0.308) |
| GloVe 6B 200d | 0.378 (−0.203) | 0.356 (−0.167) | 0.447 (−0.121) | 0.498 (−0.116) |
| $Q_{128}D_{256}S_{512}$ | 0.562 (−0.019) | 0.507 (−0.016) | 0.544 (−0.024) | 0.582 (−0.032) |
| $Q_{512}D_{1024}S_{2048}$ | 0.586 (+0.005) | 0.528 (+0.005) | 0.571 (+0.003) | 0.617 (+0.003) |

Comparing to HRED-qs, we conclude multi-task learning also helps M-NSRF achieve improved performance in query suggestion task, as the baseline employs a very similar model as ours for query suggestion. The comparison with Seq2seq with attention suggests adding attention over session information is a promising direction to further improve query suggestions quality as it emphasizes the information carried by the immediately previous query. We will pursue this in our future work.

## 4.4 ABLATION STUDY ON M-NSRF

We conducted experiments to better understand the effectiveness of different components in the M-NSRF. We also analyzed the impact of word embeddings and hidden units dimension in M-NSRF's performance. Our findings are presented in Table 4.

**Impact of Different Model Components**. To study the effect of different model components, we compared the full M-NSRF with several simpler versions of the model. At first, we turned off training for word embeddings and found a significant decrease in performance. In our training dataset, we have roughly $|OOV| = 26k$ out-of-vocabulary words and thus, training the word embeddings turned out to be very important on this data set. Also, we investigated the role of pre-trained word embeddings (ex., GloVe embeddings) and found significant performance decrease (27.9% drop in MAP) if we train M-NSRF without any pre-trained word embeddings. Hence, we can conclude that the use of pre-trained word embeddings and training them further is important to achieve better performance in M-NSRF. Second, we investigated the advantages of using max-pooling over mean-pooling and considering the last hidden recurrent state for query and document representation. We observed that max-pooling and mean-pooling provide almost the same performance while biLSTM-last approach lags slightly. To further analyze the features identified by the query and document encoders using the max-pooling technique, we followed the idea of visualization proposed in (Conneau et al., 2017). We provide an example in Figure 2 where *document 1* is clicked by the user (a positive example) and *document 2, 3* is retrieved by BM25 but not clicked (negative examples). We observed that the query and document encoders identified distinct features (e.g., the word *priceline* in the first document's title is most important) which help differentiate between clicked and unclicked documents.

In another variant of M-NSRF, we did not use the session recurrent state when computing relevance score for the candidate documents to examine the influence of previous queries from the same session on the ranking performance. We refer to this variant of M-NSRF as multi-task neural relevance model (M-NRF). From Table 4 we can find that without session information the performance drops by 2.8% in terms of MAP. We further investigated and found the session information helps particularly for longer sessions (session length $> 5$). In our evaluation dataset, we have roughly 5000 sessions of length greater than 5 (91 thousands in total).

**Impact of Dimensionality**. We further study the impact of dimensionality of the word embeddings, query, document and session latent vectors. In M-NSRF, we set the dimension of query, document and session latent vectors to 256, 512 and 1024 respectively. As shown in Table 4, decreasing the

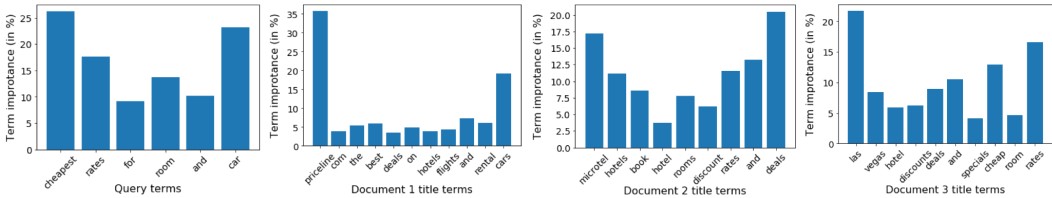

Figure 2: Example showing query and document term importance identified by M-NSRF while ranking candidate documents for the given query.

dimensions of latent vectors, decreases the performance; while increasing the dimensions further, does not affect the performance significantly. We also experimented with different dimensions of pre-trained word embeddings (50d, 100d and 200d GloVe (Pennington et al., 2014) embeddings). Word embeddings of different dimensions provide different granularity of semantic similarity; with lower dimensionality, the similarity between word embeddings might be coarse and thus hard to capture matching between two text sequences. In our experiment, we found 300 dimension for embeddings works significantly better than other dimensionality on this data set.

## 5 CONCLUSIONS

Existing deep neural models for ad-hoc retrieval often omit session information and are only trained on individual query-document pairs. In this work, we propose a context-aware multi-task neural session relevance framework which works in a sequence to sequence fashion, and show that sharing session-level latent recurrent states across document ranking and query suggestion tasks benefits each other. Our experiments and analysis not only demonstrate the effectiveness of the proposed framework, but also provide useful intuitions about the advantages of multi-task learning involving deep neural networks for two different information retrieval tasks.

As our future work, we would like to leverage the content from document body and click sequence to update M-NSRF (especially the session-level recurrent states) so that we can further explore the potential of the proposed framework for ad-hoc retrieval. As attention mechanism shows promise in improving query suggestion performance, we also also explore it in our multi-task learning setting. In addition, a broad research direction is to go beyond session boundaries to model users' long-term search goals to enhance personalized search results and query suggestions.

**Acknowledgement.** This work was supported in part by National Science Foundation Grant IIS-1760523, IIS-1553568, IIS-1618948, and the NVIDIA Hardware Grant.

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

## A    MULTITASK NEURAL SESSION RELEVANCE FRAMEWORK

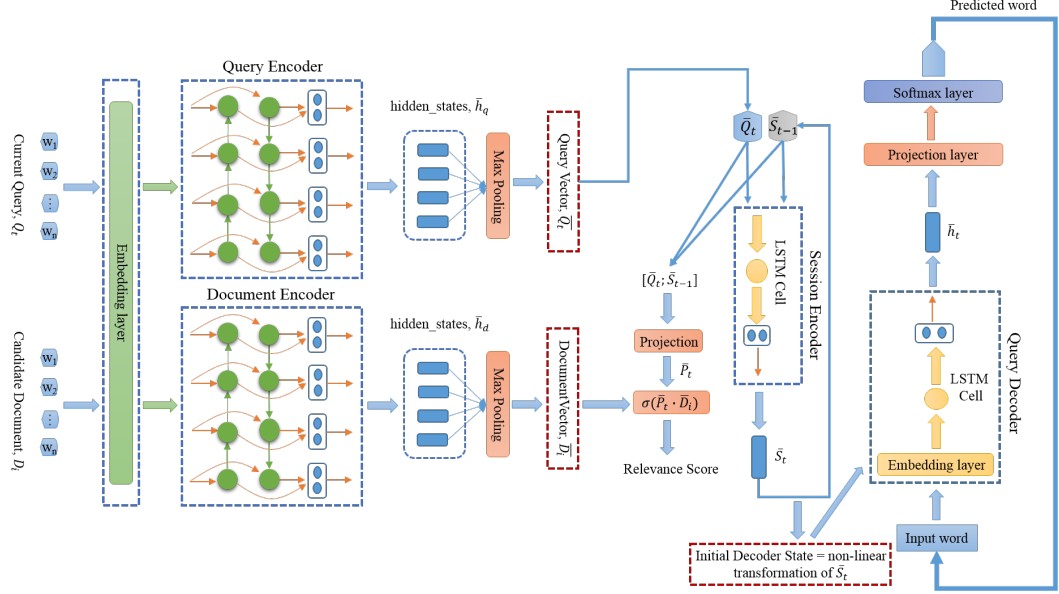

Figure 3: Architecture of the Multi-task Neural Session Relevance Framework (M-NSRF). M-NSRF uses bi-LSTM with max pooling to form query and document representations and use LSTM to gather session-level information. These recurrent states (current query representation and session-level recurrent state, which summarizes all previous queries) are used by query decoder and document ranker for predicting next query and computing relevance scores.

## B    MULTITASK MATCH-TENSOR ARCHITECTURE

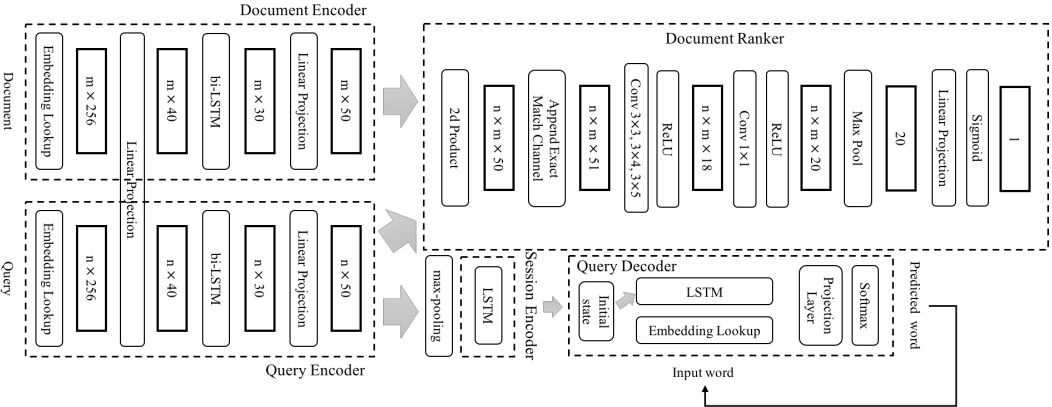

Figure 4: Architecture of the Multi-task Match-Tensor Model (M-Match-Tensor).

## C    DOCUMENT RANKING BASELINES

Deep semantic similarity model, **DSSM** (Huang et al., 2013) maps words to letter tri-grams using a word-hashing technique and uses a feed-forward neural network to build representations for both query and document. Similarity, convolutional latent semantic model, **CLSM** (Shen et al., 2014) uses word-hashing technique and uses convolutional neural networks (CNN) to build query and document representations. To compute relevance between query and document, both DSSM and

CLSM uses cosine similarity. **ARC-I** (Hu et al., 2014) uses CNN to form query and document representations and employs a multi-layer perceptron to compute relevance score. In our implementation, we used 128 convolution filters of size 1, 2 and 256 filters of size 3.

**ARC-II** (Hu et al., 2014) was proposed by focusing on learning hierarchical matching patterns from local interactions using a CNN. To keep the ARC-II model simple, we use two layers of 2d convolution and max-pooling each and two-layer feed forward neural network to compute relevance score. **DRMM** (Guo et al., 2016a) aims to perform term matching over histogram-based features ignoring the actual position of matches. In DRMM, histogram-based features are computed using exact term matching and pretrained word embeddings based cosine similarities. In principal, the histogram counts the number of word pairs at different similarity levels. The counts are combined by a feed forward network to produce final ranking scores.

The **DUET** (Mitra et al., 2017) model composed of a local and distributed model where the distributed model projects the query and the document text into an embedding space before matching, while the local model operates over an interaction matrix comparing every query term to every document term. Similarly, **Match-Tensor** (Jaech et al., 2017) model incorporates both immediate and larger contexts in a given document when comparing document to a query.

## D  QUERY SUGGESTION BASELINES

**Seq2seq** model proposed by Bahdanau et al. (2015) is a general neural network architecture that can be applied to the task where both input and output consist of a sequence of tokens. This method have been shown successful in machine translation and sequential tagging. Because different input tokens may contribute to each output token differently, attention mechanism which learns a weight between each input-output token pair can further improve the Seq2seq model. In this paper, we consider a **Seq2seq with global attention** method proposed by Luong et al. (2015), which is suitable for short text such as web queries. **HRED-qs** suggested by Sordoni et al. (2015) is very close to our work which proposed to use a hierarhical recurrent encoder-decoder approach by considering session information for context-aware query suggestion.

## E  MORE EXAMPLES OF QUERY SUGGESTION BY M-NSRF

| | |
|---|---|
| Previous session queries | discount pet supplies, homes for rent smyrna georgia |
| Next user query | homes for rent atlanta georgia |
| Suggested next query | pet friendly rentals in georgia |
| Previous session queries | language aptitude test, foreign language aptitude test |
| Next user query | american idol |
| Suggested next query | american language association |
| Previous session queries | saturday night fever, saturday night fever nj band |
| Next user query | new jersey cover band |
| Suggested next query | saturday night live |
| Previous session queries | pregnancy, abortion, abortion clinics |
| Next user query | tampa abortion |
| Suggested next query | abortion clinics in florida |
| Previous session queries | ncaa basketball, ncaa basketball trees, ncaa mens basketball bracket, sportscenter |
| Next user query | mens ncaa basketball odds |
| Suggested next query | espn |
| Previous session queries | childhood autism rating scale, childhood autism rating scale free, autism screening questionnaire |
| Next user query | pervasive developmental disorder |
| Suggested next query | how to do questionnaire |

