# OpenReview forum: "Multi-Task Learning for Document Ranking and Query Suggestion"
_ICLR.cc/2018/Conference — Accept (Poster)_

### Official Review · AnonReviewer2 · 2017-11-27
**This work proposes a multi-task learning framework, M-NSRF, that can jointly learn document ranking and query suggestion. A session recurrence is introduced and modeled by an extra LSTM. The authors carry out extensive experiments to verify their algorithm. But I think this work is not of enough novelty and the experiment design can be improved.**

**Rating:** 4
**Confidence:** 4

**Review:**

Novelty: It looks quite straightforward to combine document ranking and query suggestion.  For the model architecture, it is a standard multi-task learning framework. For the “session encoder”, it is also proposed (at least, used) in (Sordoni et al., CIKM 2015). Therefore, I think the technical novelty of the work is limited.

Clarify: The paper is in general well written. One minor suggestion is to replace Figure 1 with Figure 3, which is more intuitive.

Experiments:
1.	Why don’t you try deep LSTM models and attention mechanisms (although you mentioned them as future work)? There are many open-source tools for deep LSTM/GRU and attention models, and I see no obstacle to implement your algorithms on their top.
2.	In Table 2, M-NSRF with regularization significantly outperforms the version without regularization. This indicates that it might be the regularization that works rather than multi-task learning. For fair comparison, the regularization trick should also be applied to the baselines.
3.	For the evaluation metric of query suggestion, why not using BLEU score? At least, you should compare with the metrics used in (Sordoni et al., 2015) for fairness.
4.	The experiments are not very comprehensive – currently, there is only one experiment in the paper, from which one can hardly draw convincing conclusions.
5.	How many words are there in your documents? What is the average length of each document? You only mention that “our goal is to rank candidate documents titles……” in Page 6, 2nd paragraph. It might be quite different for long document retrieval vs. short document retrievel.
6.	How did you split the dataset into training, validation and test sets?  It seems that you used a different splitting rule from (Sordoni et al., 2015), why?

---

> ### Author Response · Authors · 2017-12-27
> **Response to the experiment related concerns of Paper250 AnonReviewer2**
>
> We thank the reviewer for providing suggestions to improve our experiments. We present the answers to the major concerns mentioned in the review.
>
> 1. The main focus of this work is to develop the multi-task learning framework for joint learning of document ranking and query suggestion via explicit modeling of user search intents in search sessions. Exploring different network architectures for document ranking and/or query suggestion is orthogonal to this work. However, as we described in Sec. 3.4, our framework is flexible and can be incorporated into other document ranker and query suggestion maker.
>
> As we demonstrated in the experiments, the proposed system is better than or competitive with the state-of-the-art approaches. Further improving its performance requires careful studies and adding model complexity may not help.  For example, we have tried to apply 2-layer LSTM as the encoder in the sequence to sequence model, but we found slight drop (~0.02) in MAP. One possible reason may be that the 2-layer LSTM has more parameters and require more data to train. However, this type of empirical study is out of the scope of this paper and we leave it for future work.
>
> 2. We test the performance of the baselines with the regularization technique and we found good improvement for the HRED-qs, 27.6/15.1/9.2/6.7. But the performance improvement for the seq2seq and seq2seq with global attention mechanism is rather marginal (~0.2%). One observation we found is that the predictive word distribution in seq2seq with attention baseline is much less skewed than ours, and since the effect of the regularizer is to reduce the skewness of the predictive word distribution, it generates less effect on the baseline than on our model.
>
> 3. As our proposed multi-task learning framework focuses on document ranking and query suggestion, our experiments mainly focused on evaluating the proposed model in these two perspectives. We compared the document ranking with a large pool of prior works and the query suggestion performance with the closest prior work, including both quantitative and qualitative comparisons, along with an ablation study. We believe the experiments presented in the paper is as comprehensive if not more than a published ICLR paper.
>
> 4. In our experimental dataset, the average length of the queries and documents (only the title field) are 3.15 and 6.77 respectively. We applied restriction on the max allowable length for query and document. We set maximum query length to 10 and maximum document length to 20. We have to admit that short document retrieval could be different from full document retrieval. As our first step towards jointly modeling the document ranking and query suggestion tasks, we follow the previous works (Huang et al., 2013, Shen et al., 2014) and limit our work to document titles only. In our future work, we will investigate the utility of our model on a full document retrieval setting.

---

> > ### Author Response · Authors · 2018-01-05
> > **New experiment results following (Sordoni et al., 2015) and analysis of the findings for query suggestion task**
> >
> > == Q3 ==
> >
> > BLEU score is widely used in sequence-to-sequence generation tasks, such as machine translation and document summarization. It measures the overlap between the generated sequence and the gold sequence. As we consider query suggestion as a sequence generation task, BLEU score is a natural choice for evaluation.
> >
> > Based on the reviewer’s insightful suggestion and avoid any potential bias in the evaluation metric, we also revised our experiment setting by following the evaluation procedure and metrics used in Sordoni et al., 2015 for the query suggestion task. However, we have to note that in Sordoni et al., 2015 they used their neural model’s output as a feature for a learning-to-rank model, rather than directly evaluating its ranking quality. In order to understand the effectiveness of these neural models in query suggestion task, we directly used the models’ output to rank the candidate queries to compare their ranking quality. The results are included in the revised paper Table 3.
> >
> > We observe that our model outperformed all baselines in this evaluation, except the Seq2seq with attention baseline. By examining the detailed suggestions resulted from these two models, we found that the attention mechanism tends to repeat the same words from the input query, and therefore is lack of variety and diversity. But our model tends to suggest highly related but totally new queries. To quantify this, we zoomed into the queries which have no overlap with their immediately previous queries, and found significant performance drop in Seq2Seq with attention baseline but significant performance improvement in our model (and our model outperformed seq2seq model by ~5%, seq2seq with attention model by ~10% and HRED-qs by 1% in this particular test set, all with p-value smaller than 0.05 in paired t-test). More details are provided in the paper. This further suggests the advantage of our model: it captures the users’ whole-session search intent and therefore suggests queries directly related their underlying intent, rather than just those immediately previous queries.
> >
> > == Q6 ==
> >
> > Due to the nature of the seq2seq, HRED-qs and M-NSRF models, their model parameters are learned from a bag of sessions. And therefore, the temporal dynamics across sessions are not considered in such models. Based on this fact, we initially randomly split the AOL search log to create our training, validation and testing sets. But we also took the reviewer’s suggestion and presented an additional experiment by following Sordoni et al., 2015  to generate background, training, development and testing dataset for conducting re-ranking based evaluation procedure to compare different models’ ranking quality in this task (see the answer to Q3). The results are reported in the revised version of our paper. As we expected, the relative performance among different models stayed the same as that in our original evaluation setup. This also suggests the validity of our original evaluation setting.

---

### Official Review · AnonReviewer3 · 2017-11-27
**I like the idea of a joint learning framework for document ranking and query suggestion, but the writing and analysis can be improved.**

**Rating:** 6
**Confidence:** 4

**Review:**

This paper presents a joint learning framework for document ranking and query suggestion. It introduces the session embeddings to capture the connections between queries in a session, and potential impact of previous queries in a session to the document ranking of the current query. I like the idea in general.

However, I have a few comments as follows:

- Multi-task Match Tensor model, which is important in the experiments (best results), is only briefly introduced in Section 3.4. It is not very clear how to extend from match tensor model to a multi-task match tensor model. This makes me feel like this paper is not self-contained. The setting for this model is not introduced either in Section 4.2.

- Section 3 is written mostly about what has been done but not why doing this. More intuition should be added to better explain the idea.

- I like the analysis about testing the impact of the different model components in Section 4.4, especially analyzing the impact of the session. It would be nice to have some real examples to see the impact of session embeddings on document ranking. One more related question is how the clicked documents of a previous query in the same session influence the document ranking of this current query? Would that be feasible to consider in this proposed framework?

- Session seems to play an important role in this multi task learning framework. This paper used the fixed 30 minute window of idle time to define a session. It would be nice to know how sensitive this model is to the definition / segmentation of sessions.

---

> ### Author Response · Authors · 2017-12-27
> **Method section and details of M-Match-Tensor model is revised to better explain the main idea.**
>
> We thank the reviewer for recognizing our contribution to multi-task learning and giving suggestions to improve the writing. We have revised our paper accordingly and respond to the major concerns here.
>
> 1. We have added a figure of the Multi-task Match-Tensor model in the appendix (see figure 4 in the revised version of the paper). We adapt multi-task learning in Match-Tensor model by adding the session encoder and the query decoder and keeping the other part of the model as it is. More details of this procedure are explained in the revised paper Section 3.4.
>
> 2. We have revised the section 3 of our paper by adding more details about the motivation and design of every component in our proposed framework.
>
> 3. We did the experiment to verify the impact of session embeddings on document ranking and result is reported in Table 4 of our revised paper. The row labeled with “M-NRF” resembles the model without considering the session embeddings in document ranking, where we observed a 2.8% drop in MAP. As the session encoder is designed to capture the search context carried till the current query, it provides an important signal for ranking documents under the current/reformulated queries.  It would be great if we could summarize what kind of queries/sessions benefit from this session recursion (i.e., where does that 2.8% drop in MAP come from).
>
> In our current model, we do not model the click sequence, and multiple clicks under the same query are assumed to be governed by the same query and session representation. As a result, the session recursion is not directly/immediately influenced by the click sequence. However, we do appreciate the great suggestion, and we believe adding another layer of session recursion at click sequence level would enable us to better capture this influence. And we will list this as a top priority in our future work.
>
> 4. We appreciate the suggestion. We followed the most commonly used threshold to define the session in IR literature. In our future work, we will study the sensitivity of the segmentation of sessions.

---

### Official Review · AnonReviewer1 · 2017-11-28
**The paper presents a multi-task learning architecture for document ranking and query suggestion, based on (predicted) user clicks.**

**Rating:** 7
**Confidence:** 4

**Review:**

The work is interesting and novel. The novelty lies not in the methods used (existing methods are used), but in the way these methods are combined to solve two problems (that so far have been treated separately in IR) simultaneously. The fitness of the proposed architecture and methodological choices to the task at hand is sufficiently argued.

The experimental evaluation is not the strongest, in terms of datasets and evaluation measures. While I understand why the AOL dataset was used, the document ranking experiments should also include runs on any of the conventional TREC datasets of documents, queries and actual (not simulated) relevance assessments. Simulating document relevance from clicks is a good enough approximation, but why not also use datasets with real human relevance assessments, especially since so many of them exist and are so easy to access?

When evaluating ranking, MAP and NDCG are indeed two popular measures. But the choice of NDCG@1,3,10 seems a bit adhoc. Why not NDCG@5? Furthermore, as the aim seems to be to assess early precision, why not also report MRR?

The paper reports that the M-NSRF query suggestion method outperforms all baselines. This is not true. Table 2 shoes that M-NSRF is best for BLEU-1/2, but not for BLEU-3/4.

Three final points:

- Out of the contributions enumerated at the end of Section 1, only the novel model and the code & data release are contributions. The rigorous comparison to soa and its detailed analysis are the necessary evaluation parts of any empirical paper.
- The conclusion states that this work provides useful intuitions about the advantages of multi-task learning involving deep neural networks for IR tasks. What are these? Where were they discussed? They should be outlined here, or referred to somehow.
- Although the writing is coherent, there are a couple of recurrent English language mistakes (e.g. missing articles). The paper should be proofread and corrected.

---

> ### Author Response · Authors · 2017-12-27
> **Addressing concerns related to the experiments**
>
> We thank the reviewer for the affirmative comments. We also appreciate the suggestions for improving the experiment comparisons and the writing quality of our paper. We have revised our paper based on the suggestions provided. We address the concerns related to the experiments as follows.
>
> 1. Our model is trained and evaluated on search sessions, which are not widely available in public datasets (as it requires continuous monitoring of a user’s search behaviors). This limited our choice in evaluation datasets. For example, most TREC tracks only consider single queries. And TREC recently has created related tracks, but with only a handful of annotated sessions (e.g., TREC Tasks Track only has 150 tasks and Session Track only has around 5 thousand sessions in total so far). Exploring more public annotated search datasets for evaluating our developed framework is definitely a very important future work of ours.
>
> 2. To help audiences easily compare our reported results with existing literature, we followed the previous works (Huang et al., 2013, Shen et al., 2014, Mitra et al., 2017) and used MAP, NDCG@1, NDCG@3, and NDCG@10 as the evaluation metrics. In this updated version, we included NDCD@5 and MRR to better compare different models' early precision, and the new results align with our previous experimental results that the proposed multi-task learning framework improves ranking performance.
>
> 3. We acknowledge that the performance of the sequence-to-sequence model with attention mechanism was better than our proposed approach in the reported average BLEU-¾. And we have revised our statement to make it more precise.

---

> > ### Author Response · Authors · 2018-01-05
> > **New experimental finding for query suggestion**
> >
> > An interesting new finding we obtained recently by paying a closer look at the query suggestion results from our method (i.e., M-NSRF) and the best baseline method (i.e., Seq2seq with attention) is that the Seq2seq with attention model tends to repeat words from the input query, which gives little variety/diversity in the suggested queries. Instead, our model is able to generate highly related but totally new queries. To verify this, we restricted the evaluation in the testing queries, which do not have overlap with corresponding input queries, and we found that M-NSRF significantly outperformed Seq2seq with attention in both BLEU scores and ranking-based metric MRR (with p-value smaller than 0.05) on such queries. In the appendix of our paper, we provided some sample query suggestion output from our model to highlight its advantage in such cases.

---

### Decision · Program_Chairs · 2018-01-29
**ICLR 2018 Conference Acceptance Decision**

**Decision:**

Accept (Poster)

**Comment:**

Overall, the committee finds this paper to be interesting, well written and proposes an end to end model for a very relevant task.  The comparisons are also interesting and well rounded.  Reviewer 2 is critical of the paper, but the committee finds the answers to the criticisms to be satisfactory.  The paper will bring value to the conference.